# Genotype–Phenotype Correlation in a Family with Brugada Syndrome Harboring the Novel p.Gln371* Nonsense Variant in the *SCN5A* Gene

**DOI:** 10.3390/ijms20225522

**Published:** 2019-11-06

**Authors:** Michelle M. Monasky, Emanuele Micaglio, Daniela Giachino, Giuseppe Ciconte, Luigi Giannelli, Emanuela T. Locati, Elisa Ramondini, Roberta Cotugno, Gabriele Vicedomini, Valeria Borrelli, Andrea Ghiroldi, Luigi Anastasia, Carlo Pappone

**Affiliations:** 1Arrhythmology Department, IRCCS Policlinico San Donato, 20097 San Donato Milanese (MI), Italy; michelle.monasky@grupposandonato.it (M.M.M.); emanuele.micaglio@grupposandonato.it (E.M.); g.ciconte@gmail.com (G.C.); giannelli.luigi@gmail.com (L.G.); emanuelateresina.locati@grupposandonato.it (E.T.L.); elisa.ramondini@grupposandonato.it (E.R.); roberta.cotugno@grupposandonato.it (R.C.); gabriele.vicedomini@grupposandonato.it (G.V.); valiborrelli91@gmail.com (V.B.); 2Department of Clinical and Biological Sciences, University of Turin, 10043 Orbassano (TO), Italy; daniela.giachino@unito.it; 3Stem Cells for Tissue Engineering Laboratory, IRCCS Policlinico San Donato, 20097 San Donato Milanese (MI), Italy; andrea.ghiroldi@gmail.com (A.G.); anastasia.luigi@hsr.it (L.A.); 4Department of Biomedical Sciences for Health, University of Milan, 20133 Milan, Italy

**Keywords:** Brugada syndrome, sudden cardiac death, genetic testing, arrhythmia, *SCN5A*, sodium channel, channelopathy, variant, mutation, humans, family, nonsense mutation, premature stop codon

## Abstract

Brugada syndrome (BrS) is marked by coved ST-segment elevation and increased risk of sudden cardiac death. The genetics of this syndrome are elusive in over half of the cases. Variants in the *SCN5A* gene are the single most common known genetic unifier, accounting for about a third of cases. Research models, such as animal models and cell lines, are limited. In the present study, we report the novel NM_198056.2:c.1111C>T (p.Gln371*) heterozygous variant in the *SCN5A* gene, as well as its segregation with BrS in a large family. The results herein suggest a pathogenic effect of this variant. Functional studies are certainly warranted to characterize the molecular effects of this variant.

## 1. Introduction

Brugada syndrome (BrS) is characterized by coved ST-segment elevation (type one BrS pattern) on the electrocardiogram (ECG), which may occur spontaneously or after administration of a sodium-channel-blocking agent such as ajmaline [1]. This syndrome is associated with an increased risk of sudden cardiac death (SCD) [1] due to ventricular arrhythmias caused by an arrhythmogenic substrate (AS) usually located in the right ventricular (RV) epicardium [2,3]. Diagnosis of BrS is challenging, since the first clinical manifestation of the syndrome may be SCD (or aborted SCD), which often occurs in early adulthood, often during sleep. Phenotypes vary greatly, even within families, and even within the same individual in response to certain factors, such as vagal tone, fever, or drugs [4].

The genetics of BrS are elusive in the majority of cases [5], with many recent studies aimed at the identification of possible new causative genes [6,7,8,9,10], while variants in the *SCN5A* gene account for about 15–30% of cases [11]. However, *SCN5A* variants are responsible for many pathologies, such as dilated cardiomyopathy, left ventricular noncompaction, arrhythmogenic right ventricular cardiomyopathy, atrial fibrillation, heart block, and various other arrhythmias [4,12,13], making genotype/phenotype predictions difficult. Therefore, recently, several studies focused on genotype–phenotype relationships involving *SCN5A* variants in BrS [4,14,15,16,17,18].

Improving the ability to molecularly confirm (or rule out) BrS is imperative, because ajmaline testing to identify the syndrome [19,20], as well as testing for inducibility of ventricular tachycardia/fibrillation (VT/VF)—which may be needed to justify the use of an implantable cardioverter-defibrillator (ICD)—often require patients to travel great distances to specialized centers [21], and the tests can be quite costly. There are also waiting lists to have these tests performed. Some patients may be physically unable to travel or too young to undergo ajmaline testing. Therefore, having a blood test that can identify the syndrome would help identify especially asymptomatic family members. However, for the majority of patients, science is currently limited, in that we need to identify other genes that may be involved and better characterize genotype–phenotype relationships for specific variants in genes that are already known to be associated with BrS.

In the present study, we report the novel NM_198056.2:c.1111C>T (p.Gln371*) heterozygous variant in the *SCN5A* gene, its segregation with BrS in a large family, as well as evidence of pathogenicity.

## 2. Case Presentations

Written informed consent of human subjects included in this case series report was obtained for their participation in the study and for publication. The procedures employed were reviewed and approved by the local ethics committee. The study was conducted in accordance with the Declaration of Helsinki, and written informed consent of human subjects was obtained for their participation in the study and for publication. The procedures employed were reviewed and approved by the local Ethics Committee (approver number: M-EC-006/A, rev. 1 March 2013).

The proband was a 39-year-old woman of Italian origin who had previously performed genetic counseling elsewhere with a positive test for the familial heterozygous variant NM_198056.2:c.1111C>T (p.Gln371*) in the *SCN5A* gene (Figure 1, LOVD: https://databases.lovd.nl/shared/variants/0000579034#00018523). She came to our attention for a second opinion because of recurrent palpitations. She underwent an ajmaline challenge at our hospital, which was positive, after which she had polymorphic ventricular tachycardia. An electrophysiological study (EPS) was performed, which was positive (Figure 2). An implantable cardioverter-defibrillator (ICD) was implanted. The proband then underwent radiofrequency catheter ablation of the AS, which was found in the epicardium of the right ventricular outflow tract. Ajmaline was administered prior to ablation to visualize the full extent of the AS (Figure 3).

### 2.1. Assessment of Family Members

The pedigree and a summary of the family history can be seen in Figure 4A.

The proband’s father (patient I-3 on family tree, Figure 4A) was a 64-year-old male with a history of at least two lipothymic episodes, bradycardia, and an implanted pacemaker before 60 years of age. He was evaluated because of this history and because of the BrS clinical diagnosis in his elder daughter (the proband). He had a positive result from an ajmaline challenge, and genetic testing was subsequently performed, demonstrating that he was a heterozygous carrier of the mutation found in the proband.

The proband’s sister (patient II-5 on Figure 4A) was a 32-year-old woman with no remarkable medical history. Due to the presence of BrS in her family, she underwent both a flecainide challenge and blood sampling for the genetic test elsewhere. Both tests were negative.

The eldest cousin of the proband (patient II-1) had a medical history characterized by sleep disturbances without sleep apnea, as well as a single syncopal episode, after which she underwent genetic counseling. Genetic testing confirmed the presence of the familial mutation in the heterozygous state with maternal inheritance. For a second opinion, she was evaluated in our hospital, where an EPS and ajmaline test were both performed, both with positive results, resulting in the implantation of an ICD.

The eldest son of patient II-1 was 14 years old. BrS was suspected due to his mother’s history of the syndrome combined with the presence of a spontaneous type two BrS ECG pattern and right bundle branch block, detected on a 12-lead ECG, in the son. For these reasons, a loop recorder was implanted. During the recordings of this device, he experienced an episode of atrioventricular nodal reentrant tachycardia (Figure 4B), sinus node arrest associated with syncope (Figure 4C), and sinus node arrest associated with dizziness and a fever of 38.3 °C (Figure 4D). For these reasons, an ICD was implanted. He underwent genetic testing on peripheral blood, which identified the familial variant in the *SCN5A* gene.

The second son of patient II-1 was 11 years old. He tested negative for the familial variant, and the family reported that he was asymptomatic to-date. The third son (from another partner) was six years old. He experienced a single syncopal episode at the age of four years. He was also affected by right bundle branch block. Due to this history, he underwent genetic testing, which demonstrated the presence of the familial heterozygous mutation in the *SCN5A* gene with maternal inheritance.

Another of the proband’s cousins, patient II-2, was a 35-year-old woman who experienced the first syncopal episode at the age of 30 years during an at-term delivery. This episode, in association with family history, prompted genetic testing from peripheral-blood-extracted DNA. This test demonstrated that she carried the familial NM_198056.2:c.1111C>T (p.Gln371*) heterozygous variant in the *SCN5A* gene. After this result, she underwent an EPS procedure elsewhere; during this procedure she experienced cardiac arrest with electrical cardioversion, followed by ICD implantation. Her sister (patient II-3) directly requested to undergo a flecainide challenge, which was negative. She also performed genetic testing with peripheral blood sampling for the familial variant in the *SCN5A* gene. This test result was negative as well.

Additional family members were reported to us by our patients, but they were followed at other centers and did not provide detailed laboratory report information to be included in this case report, nor did they consent to the use of their personal results for the purposes of publication. Thus, we have omitted this information.

### 2.2. In Silico Predictions and Variant Classification

VarSome [22] genetic database, MutationTaster, and FATHMM-MKL were used to clarify the significance of the *SCN5A* variant observed in this family. All software supported a damaging effect for the p.Gln371* variant, with VarSome reporting a pathogenic effect, MutationTaster reporting “disease causing automatic”, and FATHMM-MKL predicting “damaging”. The Genomic Evolutionary Rate Profiling (GERP) value was noted, which is defined by VarSome as “a conservation score calculated by quantifying substitution deficits across multiple alignments of orthologues using the genomes of 35 mammals. It ranges from −12.3 to 6.17, with 6.17 being the most conserved” [23]. The GERP score for the *SCN5A* variant described herein was 4.73. According to VarSome, no publications are currently listed for this variant (last accessed 4 July 2019). The DANN score, which is a pathogenicity-scoring methodology, ranging from 0 to 1, with 1 given to variants predicted to be the most damaging [24], is 0.9986. The likelihood ratio test (LRT) predicts deleterious variants through the identification of highly conserved amino acid regions using a comparative genomics data set of 32 vertebrate species, and ranges from 0 to 1 [22]. The LRT score for this variant was 0, with a deleterious prediction. Regarding the allele frequency, this variant does not have a gnomAD exomes or genomes entry, but its locus is covered in the gnomAD exomes (mean coverage = 58.2) and genomes (mean coverage = 33.9) [22].

The c.1111C>T (p.Gln371*) variant was classified as pathogenic according to criteria of the American College of Medical Genetics and Genomics and the Association for Molecular Pathology [25]:PVS1: null variant (nonsense) affecting the *SCN5A* gene, which is a known mechanism of disease (385 pathogenic variants out of 749 classified variants = 51.40%, which is greater than threshold = 10.0%), associated with Brugada syndrome, atrial fibrillation, long QT syndrome type three, idiopathic ventricular fibrillation, progressive heart block, nonprogressive heart block, sick sinus syndrome, and dilated cardiomyopathy.PM1: UniProt protein SCN5A_HUMAN intra-membrane domain “pore-forming” has seven pathogenic variants out of nine classified variants = 77.8% (greater than 66.6%).PM2: variant not found in gnomAD exomes (good gnomAD exomes coverage = 58.2). Variant not found in gnomAD genomes (good gnomAD genomes coverage = 33.9).PP3: pathogenic computational verdict because of five pathogenic predictions from DANN, GERP, LRT, MutationTaster, and FATHMM-MKL (vs. no benign predictions).

## 3. Discussion

In the present study, we report the novel NM_198056.2:c.1111C>T (p.Gln371*) heterozygous variant in the *SCN5A* gene, together with its segregation in a large family, where this variant is associated with a severe form of BrS. The pathogenicity of this variant is supported by a number of criteria, including the fact that it is found in a gene that is known to be responsible for various pathologies, as well as its extreme rarity in the general population, and the unanimous pathogenic in silico predictions. These results together strongly support a pathogenic role for this variant.

The alpha subunit of the Na_V_1.5 protein is codified by the *SCN5A* gene. Pathogenic variations in this gene result in a wide range of sodium channel dysfunctions. These variants include loss-of-function of the Na_V_1.5 protein by either decreased expression of Na_V_1.5 in the sarcolemma [26], production of non-functional channels [27], or alterations in gating properties, such as delayed activation, earlier inactivation, faster inactivation, enhanced slow inactivation, and delayed recovery of Na_V_1.5 after inactivation [28,29].

The c.1111C>T variant in the *SCN5A* gene described herein is a nonsense mutation that introduces a premature stop codon, which would result in a truncated and incomplete protein product. Previous cases involving premature stop codons in the *SCN5A* gene have also suggested a pathogenic role for nonsense variants resulting in a premature stop of the protein translation [18]. The mutation c.5464_5467del (reported as L1821fs/10) in the *SCN5A* gene has been associated with VT, recurrent aborted sudden death, and BrS [30]. The *SCN5A* missense and nonsense mutations D1430N and Q1476X resulted in a complete loss of ventricular Na^+^ current and were associated in two different families with Brugada-like ST elevation in the inferior leads or isolated conduction disturbances [31]. The *SCN5A* mutations Q1832E and R1944X have both been identified in a sudden infant death victim [32]. A frameshift mutation resulting in a premature stop codon at the C-terminus identified in a Dutch family was found to result in the translation of a non-functional channel protein that did not reach the plasma membrane, but rather was retained intracellularly [33]. Similarly, a frameshift mutation resulting in premature truncation of the protein in a 28-year-old man with monomorphic ventricular tachycardia and BrS was studied in HEK-293 cells, demonstrating reduced protein expression and the inability to record current in whole-cell patch clamp experiments [34]. These studies support the pathogenic prediction for the novel variant described in the current study and enable the diagnosis of BrS in a carrier even if a drug challenge test has not yet been performed for various reasons (patient’s age, refusal of the procedure, or travel restrictions).

Many studies have investigated the general role of the *SCN5A* gene in BrS, but with conflicting results. In one meta-analysis, BrS patients harboring an *SCN5A* mutation, with respect to BrS patients not harboring an *SCN5A* mutation, generally experienced an earlier onset of symptoms, were more likely to exhibit a spontaneous type one BrS ECG pattern, had more pronounced conduction or repolarization abnormalities, had increased atrial vulnerability, and had an increased risk of major arrhythmic events in Asian and Caucasian populations [35]. However, a standardized bioinformatics re-analysis (SIFT, PolyPhen, MutationTaster, MutationAssessor, FATHMM, GERP, phyloP, and SiPhy) and re-evaluation of frequency in the gnomAD database concluded that only a minority of *SCN5A* variants implicated in BrS fulfill the criteria for pathogenicity or likely pathogenicity based on contemporary guidelines of the American College of Medical Genetics and Genomics and the Association for Molecular Pathology [36]. These seemingly conflicting results may be explained by the grouping together of all BrS patients with *SCN5A* mutations, regardless of what type of variant was exhibited. Missense, nonsense, splicing, insertion/deletion, and frameshift mutations may all result in BrS, but alter the Na_V_1.5 protein in different ways [4,11], potentially affecting the phenotype of the patient. The importance of the Na_V_1.5 protein in BrS is clear, but better understanding the many ways in which it is altered and relating those different alterations to different severities in phenotypes could be useful.

It must be considered that a nonsense mutation can alter a channel protein more seriously than some other kinds of mutations. In fact, the Na_V_1.5 protein displays some domains that are critical for protein function; one of these is the voltage sensor domain, which is localized close to exon 9, which maps the mutation described herein [37]. Taken together, all these data provide convincing evidence that a heterozygous state for the c.1111C>T variant in the *SCN5A* gene can predispose the carrier to BrS.

Having a clear understanding of genotype–phenotype relationships can assist in the diagnosis of BrS, enabling faster, easier, and less-invasive testing, and can enable the diagnosis of patients who are unable to travel or who may refuse ajmaline testing because of fear of the procedure. Additionally, ajmaline testing is generally not performed on infants and children because of both safety and accuracy concerns. In one study, repeated ajmaline challenge after puberty unmasked Brugada syndrome in 23% of relatives with a previously negative drug test performed during childhood [38]. Therefore, ajmaline testing cannot be exclusively relied upon for the diagnosis of children. Thus, maximizing the use of genetic testing is a significant clinical goal to improve the diagnostic capabilities for all ages, but especially children and those who are unable to travel.

## 4. Concluding Remarks

The novel NM_198056.2:c.1111C>T (p.Gln371*) heterozygous variant in the *SCN5A* gene segregated with BrS in the large family presented. Taken together, the results strongly suggest a pathogenic effect of this variant and the usefulness to search for it even in patients who are asymptomatic at the time of the clinical assessment. Functional studies are certainly warranted to characterize the molecular effects of this variant, and to improve the usefulness of genetic testing to identify at-risk family members and implement preventive interventions.

## Figures and Tables

**Figure 1 ijms-20-05522-f001:**
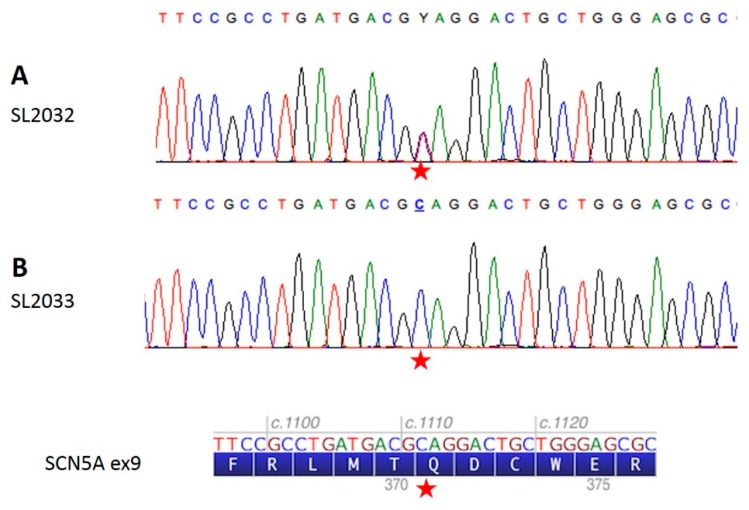
Identification of the c.1111C>T (p.Gln371*) heterozygous nonsense mutation in the *SCN5A* gene by Sanger sequencing. The red star indicates the location of the mutation (**A**) Sample SL2032 (II-4) (proband) harboring the familial variant in heterozygosis (in accordance with IUPAC nomenclature, “Y” is used as the ambiguous code to describe the simultaneous presence of both the wild type C and mutant T peaks). (**B**) Sample SL2033 (II-5) resulted negative for the mutation.

**Figure 2 ijms-20-05522-f002:**
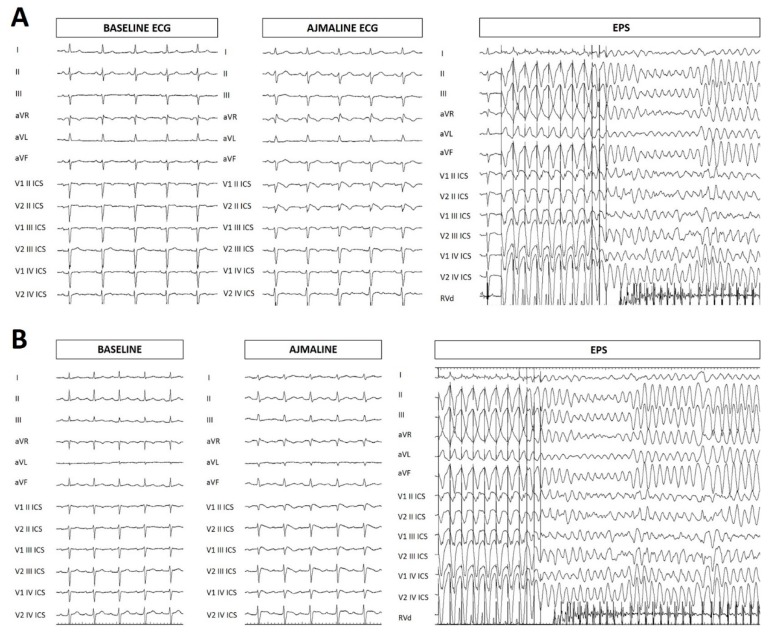
Electrocardiogram (ECG) at baseline, after ajmaline administration, ventricular tachycardia/fibrillation (VT/VF) inducibility during electrophysiological study (EPS). (**A**) Proband. (**B**) Proband’s cousin, family member II-1.

**Figure 3 ijms-20-05522-f003:**
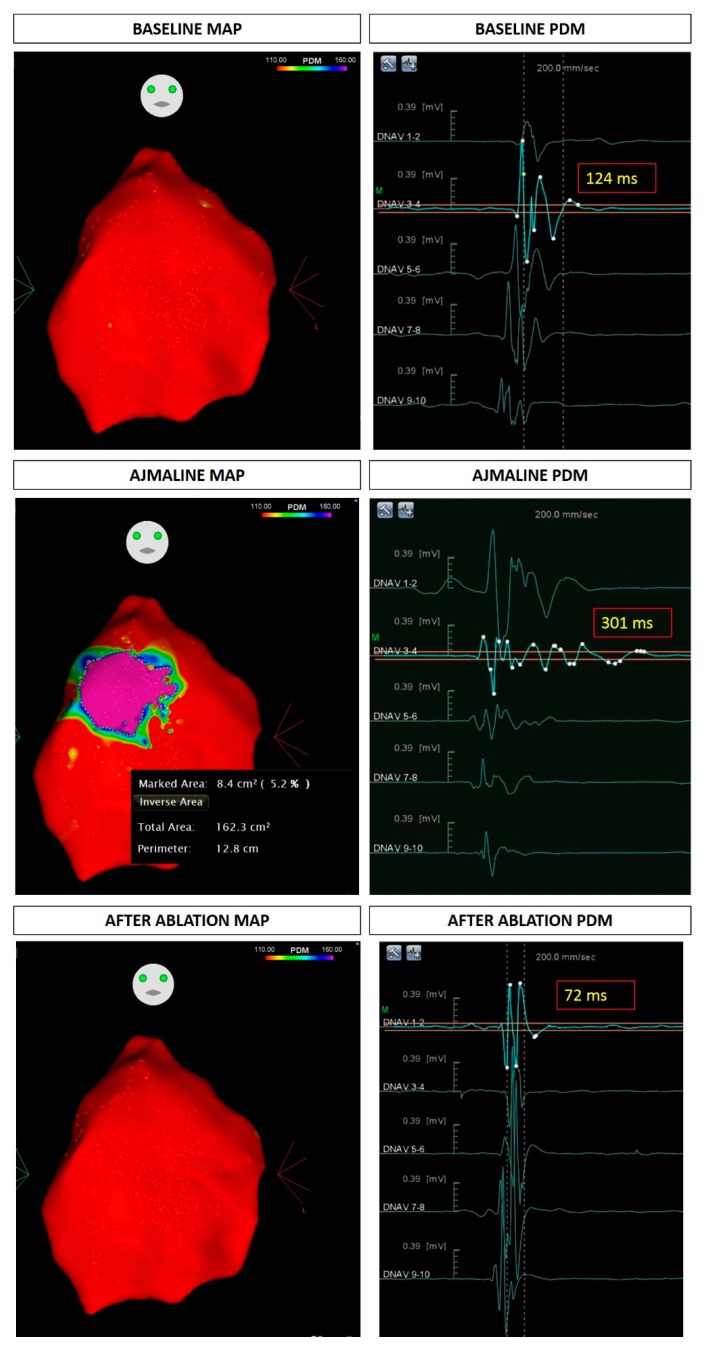
Epicardial arrhythmogenic substrate and duration of fragmented potentials in the proband at baseline, after the administration of ajmaline, and after ablation. PDM: potential duration map.

**Figure 4 ijms-20-05522-f004:**
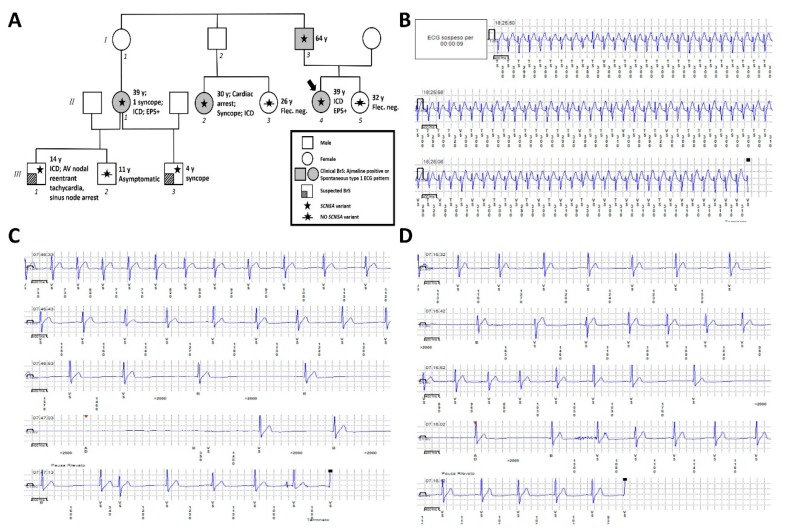
(**A**) Family pedigree. Proband identified with arrow. Square: male; circle: female; shaded background: ajmaline positive or spontaneously occurring type one Brugada syndrome (BrS) ECG pattern; lined bottom left corner: BrS clinically suspected; star: molecularly confirmed *SCN5A* variant; star with slash: genetic test for *SCN5A* variant performed but negative; y: years old at first diagnosis. (**B**) Atrioventricular nodal reentrant tachycardia of Patient III-1 detected by loop recorder. (**C**) Sinus node arrest associated with syncope in Patient III-1 detected by loop recorder. (**D**) Sinus node arrest associated with dizziness during fever in Patient III-1 detected by loop recorder.

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
