# Peer review of "Genotype–Phenotype Correlation in a Family with Brugada Syndrome Harboring the Novel p.Gln371* Nonsense Variant in the SCN5A Gene"

_ijms, 2019, doi:10.3390/ijms20225522_

Round 1

Reviewer 1 Report

This manuscript describes the pathogenic effect – in the proband and in various close family members – of a heterozygous variant in SCN5A due to a nonsense mutation introducing a premature stop codon in domain I of NaV1.5, right at the pore domain, i.e. the re-entrant extracellular loop between helices S5-S6. The pedigree analysis indicates obvious segregation with severe BrS. The truncated protein, if generated, lacks any functionality. Although extremely rare in the general population, sufficient evidence is provided to consider that any carrier of this variant will likely suffer from BrS at a certain age.

The data shown are convincing, supporting the conclusions reached. A few minor points follow:

Please include legends to all figures. This will make the text easier to follow, particularly when symbols are used. For instance, in Fig.1, why is “Y” shown instead of “T” to indicate the replaced nucleotide? Also, regarding Fig. 4, patient II-2 is referred to in the text as a 35-year-old woman (pg. 6, line 3), whereas it says she is 30 in the pedigree of panel a. Section 2.2. may require some rearrangement. Thus, wouldn’t it be better to define first the ACMG criteria, and next, the pathogenicity expected based on various in silico comparisons performed? In pg. 7, line 22, please replace “left in the cytosol” by “retained intracellularly”. The cytosol is the intracellular fluid and, in the paper referred (Ref. 33), NaV5 truncated at the S5-S6 loop of domain IV is likely inserted properly in the endoplasmic reticulum membrane, yet then it gets stuck, failing to be delivered to the cell surface. Finally, the manuscript would benefit from some corrections for English grammar. For instance, use "before 60 years of age", "before 60 years old", or better "before the age of 60" (pg. 5, line 4), and analogously whenever referring to individuals’ age.

Author Response

Thank you for your very positive review. The figure legends were submitted, then deleted by the journal during the typesetting stage. I have contacted the journal, and they have said that they would fix this. When we upload the response to the reviewers, we will upload the Word document created by the authors, rather than the typeset version. In Figure 1, panel A reports the Sanger chromatogram obtained from the proband DNA (DNA code SL2032). The Y letter indicates the heterozygous peak (concomitant wild type C allele and mutant T allele) in accordance with IUPAC nomenclature. This comment is now included in the legend of Figure 1. The ages in Figure 4 refer to the age at first diagnosis, rather than the current age. This is now clarified in the figure legend.  “Left in the cytosol” has now been replaced by “retained intracellularly”. The English grammar has been checked, especially in reference to the age of the individuals. We thank you again for your time and your positive review.

Reviewer 2 Report

This is a report of a family with a novel variant in SCN5a presenting with Brugada but variable phenotype within the family, consistent with previous reports of SCN5A variant behaviour. It is well presented and well written with clear and helpful illustrations. Generally speaking, there is a lack of such detailed family reports of phenotype-genotype cosegregation to validate the pathogenicity of such novel variants, so this paper adds usefully to the literature and will be helpful for others who find this variant subsequently. I have no important comments, the paper is well put together and is a good illustration of nonsense SCN5A variant behaviour. Please clarify what is meant by a lipothymic episode.

Author Response

Thank you for your very positive review. A lipothymic episode is a feeling of faintness without loss of consciousness. We thank you again for your time and your positive review.

Reviewer 3 Report

The authors report a novel nonsense SCN5A mutation that causes Brugada syndrome (BrS) in an Italian family. BrS is generally difficult to diagnose and genetic tests can help to identify patients, especially the asymptomatic ones.

Specifics comments:

Which other genes have been screened? Any other mutations or low frequency variants found? Where was the arrhythmogenic substrate found? (RV epicardial?) Did any of the family members BrS positive have a long QT interval? Possible overlap syndrome? Any electrical storm reported after ICD implantation? Follow up after ablation and ICD implantation?

Main structure comment:

Figure legends are missing

Figure 3 (with the  Potential duration maps) is cutted.

Author Response

Thank you for your very positive review. The genetic analysis of this family was conducted with a "one gene at a time" approach. Thus, only the SCN5A gene was analyzed in the proband DNA and only the mutation was investigated in the relatives. The arrhythmogenic substrate was found in the epicardium of the right ventricular outflow tract, and this information has now been added to the description of the proband in the Case Presentation section. No long QT interval or suspicion of an overlap syndrome was detected in any patient. No events in patients who consented to publication were detected after ICD implantation, other than those already reported in the manuscript. No electrical storms were detected. Only the proband both had ablation at our center and agreed to be included in the publication: this patient has not had any events after ablation to date. The figure legends were submitted, then deleted by the journal during the typesetting stage. I have contacted the journal, and they have said that they would fix this. When we upload the response to the reviewers, we will upload the Word document created by the authors, rather than the typeset version. We thank you again for your time and your positive review.